# Knowledge Level and Hand Hygiene Practice of Nepalese Immigrants and Their Host Country Population: A Comparative Study

**DOI:** 10.3390/ijerph17114019

**Published:** 2020-06-05

**Authors:** Lorna K.P. Suen, Tika Rana

**Affiliations:** 1School of Nursing, The Hong Kong Polytechnic University, HungHom, Kowloon GH506, Hong Kong; 2The Nethersole School of Nursing, The Chinese University of Hong Kong, Ma Liu Shui 6/F, Hong Kong; rana1234t@gmail.com

**Keywords:** epidemiology, public health, infection control, Nepalese, hand hygiene

## Abstract

Nepali people are one of the ethnic minority immigrants of Hong Kong. This epidemiological investigation aims to determine and compare the knowledge level and hand hygiene (HH) behaviour of the Nepali people and the population of their host country (i.e., native Chinese population of Hong Kong). A total of 1008 questionnaires were collected via an online platform. The overall knowledge level of the native population towards HH was considerably higher than that of the Nepali respondents. Lower levels of knowledge in young and older people were noted. Reduced HH knowledge was also observed in people with low educational level or with comorbid illness(es). Significant differences between groups were noted in the self-reported hand washing behaviours. Regarding hand drying, more Nepalese than native Chinese respondents always/sometimes dried their hands on their clothing irrespective whether after performing handwashing in public washrooms or at home. Misconceptions and suboptimal practices on HH were prevalent in the two populations. The findings of this comparative study offer valuable information for the development of culturally sensitive health educational programs to enhance HH practices for the ethnic minorities and native Chinese population.

## 1. Introduction

Handwashing is a simple and cost-effective method for preventing diarrhoea and respiratory illnesses in communities and institutions [1]. Inadequate sanitation facilities and poor knowledge about hand hygiene (HH) are associated with increased risk of contracting infectious diseases [2].

Nepal, a landlocked country in South Asia, has more than 1 million people who still practice open defecation. Approximately 15% of households in the country have no toilet facilities (21% and 11% in rural and urban areas, respectively), and 20% have no supplies of water, soap or any other available hand cleaning agents [3]. The handwashing behaviours of people residing in slums are generally suboptimal. Only 20% of mothers in Nepal wash their hands with soap after using the toilet and 14% wash their hands with soap after cleaning their babies’ bottoms. Handwashing before cooking or feeding the baby is rarely performed [4]. Infectious diseases, such as influenza, malaria, pneumonia and diarrhoea, are the top leading causes of death in Nepal [5]. Numerous factors, such as poor HH, unsafe water and inadequate sanitation facilities, as well as lack of infection control experts and specialised laboratories, are associated with high prevalence of infectious diseases in Nepal [6]. Although sanitation facilities have substantially improved in recent years [7], the common standard of HH practices in the mother country (Nepal) that is deep-rooted in the Nepali population has not greatly changed and even continued to be observed by Nepali immigrants. 

Hong Kong is a multicultural global city because of transnational immigrations. In Hong Kong, the history of Nepali people, who account for roughly 8% of the total ethnic minority populations, is particularly linked to the development of the military requirements of Hong Kong [8]. These people primarily moved to the territory as part of the Gurkha Brigade of the British Army in the 1960s. Their primary duties were to combat illegal immigrants and protect the population of Hong Kong. After the handover in 1997, the British government withdrew from Hong Kong, and the Gurkha brigade was disbanded. Some Nepalese settled in Hong Kong, and others stayed because their children were born in this host country (Hong Kong) [8,9,10]. Early South Asian settlers minimally interacted with local Chinese because of language barrier and cultural differences [9]. Compared with the local population in Hong Kong, more than 80% of the Nepali people in Hong Kong are working in elementary occupation and at lower education levels, longer working hours, less stable jobs and lower salaries [8]. These South Asian immigrants are subject to consequent ignorance and negligence at every level of the society and are at the most disadvantaged position because of their lower socioeconomic status [11].

Immigrants from high endemic areas are at greater risk of developing infectious diseases than the people in their host country [12]. Nepalese in Hong Kong are at a higher risk of developing various infectious diseases than the population in Hong Kong. Therefore, the likelihood of transferring these diseases from the Nepalese population to the people in Hong Kong increases.

Although the Nepalese have been living in Hong Kong for over five decades, less is known about the HH knowledge level and behaviours among the Nepalese [13]. No study has examined the HH knowledge level and practices among the Nepalese and the native Chinese population of Hong Kong. The findings of this study can contribute to the existing body of knowledge about the similarities and differences in HH practices between these two populations. Moreover, the study can provide valuable information to health care professionals developing culturally sensitive programs for enhancing HH practices. This strategy can eventually reduce the risk of infectious diseases in both groups who are living and interacting closely in the same society.

HH efficiency is a combination of washing efficiency and hand drying. Hand hygiene practices are recommended as one of the effective preventive measures to prevent the transmission of the COVID-19 virus [14]. Empirical evidence indicated that handwashing is approximately 85% effective in removing microorganisms on hands, and hand drying provides a further reduction in transient flora [15]. All these considered, the handwashing and hand drying practices of the two populations were explored and compared in this study.

### Aim of the Study

This epidemiological investigation aims to determine and compare the HH knowledge level and behaviour of the Nepali people and the native Chinese population of Hong Kong. Handwashing and hand drying behaviours were investigated. The findings of this cross-sectional study substantially contribute to the understanding of the knowledge gap and the HH behaviour of the two populations. Subsequently, the findings provide information to deliver culturally sensitive health promotion activities and campaigns for the improvement of HH compliance.

## 2. Materials and Methods

### 2.1. Subjects and Procedure

Nepali immigrants and the native Chinese population of Hong Kong who were 18 years or above and resided in Hong Kong during the survey were recruited by network sampling. SurveyMonkey was used as a platform to collect data [16]. This online survey application can distribute the questionnaire by email, smartphones via applications, such as WhatsApp, and social media platforms, such as Facebook. SurveyMonkey allows participants to access the questionnaire, guides respondents to complete all the items, conducts analyses and exports results [17]. As most elders may not be smartphone users, questionnaires with a paper-and-pen format were distributed to a small number of participants, such as elders, to ensure good coverage of respondents and increase the generalisability of the findings. 

The protocol of this study was approved by the Ethics Committee of the participated university (HSEARS20180409006). Participation to this survey was voluntary, and the confidentiality of the data was strictly observed. Participants were fully aware of the purpose of the study before proceeding with the online-based survey. The institutional review board of our university approved the form of consent indicated by completing the survey.

### 2.2. Instrument

The handwashing and drying questionnaire was constructed according to the literature related to handwashing and drying [18,19,20,21,22,23,24]. This questionnaire consisted of three parts: Part 1 mainly collected the sociodemographic data and personal habits (14 items). Part 2 focused on knowledge on HH (12 items) by using true or false responses. The items were related to common myths and fallacies of HH reported in the literature. The score ranged from 0 to 12, and a high score indicated a high knowledge level on HH. Part 3 comprised items related to self-reported handwashing and hand-drying practices, including but not limited to the most common handwashing methods under 14 specific conditions, usual duration for lathering the hands with soap before rinsing and hand drying methods commonly used in public washrooms and at home. The validity and reliability of the questionnaire were reported in detail in another paper [25]. Given the scope of this paper, only some items in the original questionnaire were reported here.

### 2.3. Data Analysis

Descriptive statistics was used to present the sociodemographic characteristics, knowledge level on HH and self-reported HH behaviours of the respondents. The association between categorical variables was examined with *x^2^* test or Fisher’s exact test. Independent t-test or one-way ANOVA was used to determine the group differences in the knowledge score on HH. SPSS version 25.0 (IBM Corporation, Armonk, NY, USA) was used for all statistical analyses. All statistical tests were two-sided, with significance level set as *p <* 0.05.

## 3. Results

Data were collected from January 2018 to February 2019. A total of 1008 questionnaires were collected (482 from Nepalese residing in Hong Kong and 526 from the native Chinese population of Hong Kong). The format for completing the questionnaire included the use of SurveyMonkey (*n* = 902, 89.5%), self-administered paper-and-pen format (*n* = 14, 1.4%) or assisted paper-and-pen format (*n* = 92, 19.0%). The average estimated time to complete the questionnaire on SurveyMonkey of the Nepalese and the Chinese population were 16 and 11 min, respectively, as recorded by the online platform.

### 3.1. Sociodemographic Characteristics of Respondents

The respondents between the two populations were fairly distributed in age group (18–29, 30–49, 50–59, and 60 years or above) and working status. By contrast, statistically significant differences were observed in certain sociodemographic characteristics of respondents between groups. These differences were reflected by the high percentages of Nepali respondents who were males (56.2%), married/partnered (71.6%), with primary or below educational level (20.3%), without influenza vaccination over the past 12 months (91.9%) and with a habit of wearing rings (35.5%), artificial/acrylic nails (35.5%) and bracelets (51.7%). The habit of the local population to wear watches (58.6%) was higher. Exactly 60% of Nepalese used bare hands for eating, and 27% expressed they have currently stopped this previous habit (Table 1).

### 3.2. Knowledge Level towards HH

The knowledge level of the Chinese population towards HH was considerably higher than that of the Nepali respondents (9.55 vs. 7.12 out of 12). A considerable number of Nepali respondents could not differentiate the diseases that can or cannot be transmitted with poor HH, especially the item related to the human immunodeficiency virus infection (Nepalese vs. local: 47.7% vs. 2.7%), and upper respiratory tract infection/influenza-like illness (Nepalese vs. local: 56.6% vs. 85.9%). The concepts associated with HH among the Nepalese were relatively poor. Almost half of the respondents carried the misconceptions that always keeping the hands clean may decrease the body’s defence mechanism (49.4%, *n* = 238), hands must be held under water while lathering (48.1%, *n* = 232), an alcohol-based hand sanitiser containing 40% alcohol was sufficient for disinfection (41.4%, *n* = 199) and that lathering the hands for 10 s before rinsing was enough for disinfection (58.0, *n* = 279). Over 30% of the respondents believed that water temperature caused a difference in the hand cleaning effect (37.8%, *n* = 133) (Table 2).

There are significant age-related differences, with lower levels of knowledge in young and elder people. Reduced HH knowledge was also observed in people with low educational level (primary or below) or with comorbid illness(es). Respondents with better HH knowledge generally performed HH more often during infectious disease outbreaks and took longer time to lather the hands with soap before rinsing. Females exhibited a significantly higher HH knowledge than males in the Chinese respondents (*p* = 0.018), but this gender difference was not observed in the Nepali respondents (Table 3).

### 3.3. Self-Reported Hand Cleaning and Hand Drying Practices

Significant differences in the self-reported hand washing behaviours were seen between groups. For example, more Nepalese than the Chinese respondents indicated that they did not perform hand washing after handling food, cooking, urinating, defalcating, caring for a sick person, showing visibly dirty hands, touching livestock or animal waste, gardening or disposing garbage bag. Over 40% of the Nepalese indicated that they only hand wash for ≤ 5 s. An increased number of local respondents would hand wash more often during infectious disease outbreaks than the Nepalese (77.0% vs. 45.9%), with the latter group inclined to ignore hand washing if they are in a hurry (*p* < 0.001), nobody is in the washroom (*p* < 0.001) or they only urinated (*p* < 0.001).

Regarding hand drying, more Nepalese always/sometimes dried their hands on their clothing (92.7% vs. 42.6%, *p* < 0.001) or used personal towel or handkerchief (83.0 vs. 35.8%, *p* < 0.001) than the Chinese respondents. The Chinese respondents generally preferred using paper towels supplied by the washrooms than the Nepalese (96.7% vs. 65.8%). Respondents might shake their hands to remove excess water before drying and limit the use of paper towels to two pieces. The average time for using either warm or jet hand dryers in the two groups were generally less than 20 s. For the usual hand drying practice at home, the Nepalese preferred to rub their hands on their clothes or by air evaporation, whereas the Chinese population preferred using a towel shared with family members (Table 4).

## 4. Discussion

This study is the first to examine comparatively the HH knowledge level and behaviours among the Nepalese and the Chinese population of Hong Kong. Many Nepali respondents exhibited a habit of wearing ring(s) and artificial/acrylic nails or bracelets, whereas the local respondents were predominantly watch wearers. Long nails were associated with the presence of enteric bacteria, whereas ringed fingers were associated with increased bacterial load [26]. Therefore, special attention to these regions during hands washing should be made.

A considerable number of Nepali respondents indicated that they use bare hands for eating. Eating using the hands is a common practice in many countries, such as Nepal, India, Africa and the Middle East. Eating using the hands is not only seen as a social politeness in these countries but is believed to increase blood circulation, promote a sense of fullness and satiety and heighten sensual connection with food during eating [27,28]. Therefore, the importance of performing HH before eating must be emphasised, and the fingernails must always be kept short and clean. Approximately 27% of Nepali respondents expressed that they stopped using their hands for eating, which was probably due to acculturation after residing in Hong Kong. A very high percentage of Nepoli respondents did not receive influenza vaccination over the past 12 months (91.9%). In an online survey conducted by Stedman-Smith and her team [29], it has been reported that the influenza vaccine uptake was associated with protective hand hygiene behaviours. This may be due to a conscious effort of the respondents to practice the two behaviours together as a more thorough approach for preventing infectious disease.

The knowledge level of the Chinese population towards HH was significantly higher than that of the Nepali respondents. A considerable number of the Nepalese could not differentiate the diseases that can or cannot be transmitted with poor HH, especially in the item related to human immunodeficiency virus infection. This finding may be due to the relatively low educational level of this minority group. Respondents with good HH knowledge in the two groups were associated with positive HH behaviours, such as performing HH often during infectious disease outbreak and increasing lathering time of the hands with soap before rinsing.

Another epidemiological investigation of Hong Kong conducted by the research team [25] revealed that being female, middle-aged and with tertiary education level are protective factors to improve HH knowledge. In this study, females showed significantly higher HH knowledge than males in the Chinese respondents, but this gender difference was not observed in the Nepali respondents. This finding may be because female Nepali immigrants were usually assigned a passive mother–wife role and achieved relatively lower educational level than men. Thus, a lesser dominant role than the females in Hong Kong was observed [7,10].

More Nepalese indicated that they do not wash their hands after handling food or cooking, urinating or defecating, caring for a sick person, possessing visibly dirty hands, touching livestock or animal waste, gardening or disposing of garbage bag than the Chinese respondents. Over 40% of the Nepalese indicated that they only handwash for ≤5 s. A considerable number of Nepalese respondents were inclined to ignore handwashing if they are in a hurry, if nobody is in the washroom or if they only urinated. According to Tonsing [13], immigration involves an array of changes in people’s lives. Such changes can sometimes be overwhelming as individuals encounter various changes that affect them economically, culturally and psychologically and impede their immigration experiences. Huge differences in the infrastructure between Hong Kong and Nepal have been observed. The former has been classified as a high-income country, and the latter as one of the lowest income and least developed countries by the United Nations [30]. Previous handwashing behaviours, which have been developed in the mother country (Nepal) and are deeply rooted in the Nepalese, may not be easily changed even if they moved to other host countries. This practice may be especially difficult for the old generation who originated in Nepal slums who experienced defecating in public, in which no proper washing and drying facilities are available [3]. Thus, these older people may find hard adapting to another culture and practice that greatly contrasts with their practices. For example, washrooms may be housed with advanced sanitation facilities, such as hand drying devices, hands-free faucet with motion sensor and doors with automatic control, in the host country (Hong Kong).

People who spoke the primary language of the country and residence duration in a country were commonly used as indicators to determine acculturation levels [31]. HH is considered a social norm that is an effective driver to follow other’s behaviour in a relevant social group [32,33]. Lengthened residency in a country implies wider exposure and interaction with the Chinese population. This situation leads to increased potential to adopt the values and beliefs of the Chinese and follow their general behavioural patterns [31] as well as tendency to practice socially acceptable behaviour.

Regarding hand drying, more Nepalese than the Chinese respondents always/sometimes dried their hands using own clothing irrespective whether after performing handwashing in public washrooms or at home. Drying hands on dirty clothes can compromise the benefits of handwashing [34]. Hands that are inadequately dried are more likely to transmit microorganisms compared with hands that have been completely dried [35].

Nepalese in Hong Kong are at a higher risk of developing various infectious diseases than the Chinese population in Hong Kong. The likelihood of transferring the infectious diseases from the Nepalese population to others residing in the same community is high as well. The incidence rate of tuberculosis infection is comparatively high among all the immigrants and minority population living in than people born in the UK [36]. Immigrants from underdeveloped countries with high endemic areas show high risk of acquiring similar infectious diseases in the immigrated areas due to poor HH compliance [12]. The incidence rate of infectious diseases is comparatively high among the immigrant workers from South Asian countries than the general Singaporean population in their host country [37]. Therefore, improving the HH practices of the Nepali population residing in Hong Kong can eventually decrease the risk of infectious diseases for the minor population and population at large.

Nepalese are one of the fastest growing ethnic minorities in Hong Kong [38], and they have considerable visibility in the society. Nepalese are organised and have established their organisations to fight for their rights [39]. Educational talks and promotional activities on HH can be effectively delivered via these bodies to reach out to these people.

Misconceptions and suboptimal practices on HH were not only prevalent among Nepalese but also among the Chinese population in Hong Kong. For example, majority of the respondents in both groups misunderstood holding the hands under water must be done while lathering, using 40% alcohol and alcohol-based hand-rub (ABHR) is sufficient for disinfection, lathering for 10 s before rinsing is enough for disinfection and water temperature may cause a difference in cleaning effects. Handwashing time and degree of friction generated during lathering are more important than water temperature in removing dirt and microorganisms, and warm water causes skin irritations and is not environmentally friendly [40,41]. The average times for using either warm or jet hand dryers in the two groups were generally less than 20 s. As a result, a considerable excess amount of water on the hands may easily re-contaminate the hands after touching the surface environment, such as door handles upon leaving washrooms. Although the majority of the Chinese respondents indicated that they clean their hands under different specific situations, they admitted that they only use water or, in rare occasions, ABHR or wet wipes for hand cleaning instead of handwashing with soap. By breaking the chain of infection and reducing the incidence of infection, hands must be washed correctly and at the right time [42]. Proper education of the public is necessary to fill these knowledge gaps.

The findings of this comparative study shed light on the understanding of the knowledge level and behaviour towards HH of the Nepalese and the Chinese population of Hong Kong. This study offers valuable information for the development of culturally-sensitive health promotional programs to enhance HH practices for the ethnic minorities and general population at large.

### Limitations and Recommendations

This study exhibits a few limitations. HH is considered a socially acceptable behaviour. Thus, respondents may over-report the situation. The non-obtrusive monitoring of HH behaviour may be considered in future studies to provide an unbiased evaluation of the actual behaviour. Variables, such as duration of residency in the host country and place of birth, must be collected in the future survey to understand how these factors can influence one’s acculturation level and behavioural patterns. Future studies must compare the knowledge and HH behaviours between the Nepalese in their home country and those immigrants to other countries. Furthermore, studies must investigate the change in the behavioural patterns before and after substantial improvement in the sanitation facilities at Nepal in recent years. Research must be conducted to facilitate our understanding of the environmental impact on HH behaviours. Inclusion of other common ethnic group, such as Filipinos or Indonesians, in future studies can provide valuable information for the development of culturally sensitive interventions related to HH in Hong Kong.

## 5. Conclusions

The Chinese population of Hong Kong generally exhibited better knowledge level and more favourable HH behaviour than the ethnic minorities from Nepal. Misconceptions related to the concepts associated with HH were prevalent in the two populations, indicating the need for educational enhancement. The findings of this comparative study offer valuable information for the development of culturally sensitive health educational programs to enhance HH practices for the ethnic minorities and the general population of Hong Kong. 

## Figures and Tables

**Table 1 ijerph-17-04019-t001:** Sociodemographic characteristics of respondents.

	Nepalese Residing in Hong Kong (*n* = 482)	Native Chinese Population of Hong Kong (*n* = 526)	Test Statistics & *p*-Value
Variables	*n* (%)	*n* (%)	
Age group			
18–19 20–29 30–39 40–49 50–59 60 or above	24 (5.0) 115 (23.8) 128 (26.6) 96 (19.9) 78 (16.2) 41 (8.5)	21 (4.0) 119 (22.6) 130 (24.7) 97 (18.4) 96 (18.3) 63 (12.0)	0.429 ★
Gender			
Male Female	271 (56.2) 211 (43.8)	217 (41.3) 309 (58.7)	< 0.001 ★ ***
Marital status			
Single Married/Partnered Widowed/Separated/Divorced	110 (22.8) 345 (71.6) 27 (5.6)	233 (44.3) 271 (51.5) 22 (4.2)	< 0.001 ★ ***
Educational level			
Primary or below Secondary Tertiary /College or above	98 (20.3) 149 (30.9) 235 (48.8)	5 (1.0) 127 (24.1) 394 (74.9)	< 0.001 ★ ***
Working status			
Full time Part time Nil ✡	268 (55.6) 29 (6.0) 185 (38.4)	294 (55.9) 27 (5.1) 205 (39.0)	0.827 ★
Comorbid illness			
No Yes	398 (82.6) 84 (17.4)	425 (80.8) 101 (19.2)	0.467 ★
Self-rated health condition			
Excellent Good Average Fair Poor	52 (10.8) 295 (61.2) 40 (8.3) 28 (5.8) 67 (13.9)	55 (10.5) 257 (48.8) 174 (33.1) 36 (6.8) 4 (0.8)	< 0.001 ★ ***
Receive influenza vaccination over the past 12 months			
No Yes	443 (91.9) 39 (8.1)	414 (78.7) 112 (21.3)	< 0.001 ★ ***
Have a habit of wearing ring(s)			
No Yes	311 (64.5) 171 (35.5)	415 (78.9) 111 (21.1)	< 0.001 ★ ***
Have a habit of wearing artificial/acrylic nails			
No Yes	312 (64.7) 170 (35.3)	496 (94.3) 30 (5.7)	< 0.001 ★ ***
Have a habit of wearing watch			
No Yes	310 (64.3) 172 (35.7)	218 (41.4) 308 (58.6)	< 0.001 ★ ***
Have a habit of wearing bracelet			
No Yes	233 (48.3) 249 (51.7)	461 (87.6) 65 (12.4)	< 0.001 ★ ***
Use bare hands for eating without using eating utensils			
No Previously yes, but now not Yes	83 (13.0) 130 (27.0) 289 (60.0)	436 (82.9) 21 (4.0) 69 (13.1)	< 0.001 ★ ***
Format for completing the questionnaire			
Via ‘Survey Monkey’ Paper and pen (self-administered) Paper and pen (with assistance)	395 (82.0) 0 (0.0) 87 (18.0)	507 (96.4) 14 (2.6) 5 (1.0)	< 0.001 ◈ ***

Note: ◈—Fisher’s exact test; ★—Chi-square test; ***—Statistically; significant at *p* < 0.001; ✡—retired/unemployed/housewife/student/voluntary job.

**Table 2 ijerph-17-04019-t002:** Knowledge level on hand hygiene (0 to 12).

Which of the Following Diseases Can Be Transmitted by Poor Hand Hygiene?				
		Nepalese Residing in Hong Kong (*n* = 482)	Native Chinese Population of Hong Kong (*n* = 526)	Test Statistics & *p*-Value
Item	Diseases	*n* (%)	*n* (%)	
1	Diarrheal disease			
	True ϕ False	424 (88.0) 58 (12.0)	481 (91.4) 45 (8.6)	0.069 ★
2	Upper respiratory tract infection/ Influenza-like illness			
	True ϕ False	273 (56.6) 209 (43.4)	452 (85.9) 74 (14.1)	< 0.001 ★ ***
3	Hand-foot-mouth			
	True ϕ False	348 (72.2) 134 (27.8)	510 (97.0) 16 (3.0)	< 0.001 ★ ***
4	Human immunodeficiency virus			
	True False ϕ	229 (47.7) 251 (52.3)	14 (2.7) 512 (97.3)	< 0.001 ★ ***
5	Skin ulcer			
	True ϕ False	276 (57.4) 205 (42.6)	389 (74.0) 137 (26.0)	< 0.001 ★ ***
6	Eye infections			
	True ϕ False	362 (75.4) 118 (24.6)	512 (97.3) 14 (2.7)	< 0.001★***
7	Diabetes			
	True False ϕ	151 (31.3) 331 (68.7)	1 (0.2) 525 (99.8)	< 0.001◈***
**Are the following statements correct?**				
8	Always keeping your hands clean may lower our body defence mechanism			
	True False ϕ	244 (50.6) 238 (49.4)	95 (18.1) 431 (91.9)	< 0.001★***
9	Hands should be held under water while lathering with soap			
	True False ϕ	250 (51.9) 232 (48.1)	188 (35.7) 338 (64.3)	< 0.001 ★ ***
10	An alcohol-based hand sanitizer that contain 40% alcohol is sufficient for hands disinfectant			
	True False ϕ (Answer should be 60%)	283 (58.7) 199 (41.3)	239 (45.4) 287 (54.6)	< 0.001 ★ ***
11	Rubbing my hands until soap forms a lather for 10 seconds before rinsing is enough for hand disinfection			
	True False ϕ (Answer should be 20 seconds)	202 (42.0) 279 (58.0)	310 (58.9) 216 (41.1)	< 0.001 ★ ***
12	Temperature of water makes no difference in terms of the cleansing effect of hand cleaning			
	True ϕ False	219 (62.2) 133 (37.8)	370 (70.3) 156 (29.7)	0.012 ★ *
13	Total number of correct			
	(0–12) mean ± standard deviation	7.12 (2.41)	9.55 (1.58)	< 0.001 ▲ ***

Note: ϕ—correct answer; ◈—Fisher’s exact test; ★—Chi-square test; ▲—Independent *t*-test; *—Statistically significant at *p* < 0.05; ***—Statistically significant at *p* < 0.001.

**Table 3 ijerph-17-04019-t003:** Association between knowledge level on hand hygiene (0 to 12) and characteristics of participants.

Variables	Nepalese Residing in Hong Kong (*n* = 482)	F Statistics ▲ & *p*-Value	Native Chinese Population of Hong Kong (*n* = 526)	F Statistics ▲ & *p*-Value
	Mean (SD)		Mean (SD)	
Age group				
18–19 20–29 30–39 40–49 50–59 60 or above Total:	6.33 (2.22) 6.47 (2.49) 7.48 (2.41) 7.75 (2.56) 7.50 (1.89) 6.07 (2.02) 7.12 (2.41)	6.36 < 0.001 ***	8.86 (0.79) 9.72 (1.46) 9.87 (1.36) 9.55 (1.58) 9.57 (1.52) 8.76 (2.14) 9.55 (158)	5.52 < 0.001 ***
Gender				
Male Female	7.13 (2.42) 7.11 (2.41)	0.25 0.928	9.35 (1.60) 9.69 (1.55)	1.11 0.018 *
Marital status				
Single Married/Partnered Widowed/Separated/Divorced	6.75 (2.44) 7.29 (2.39) 6.41 (2.33)	3.37 0.035*	9.61 (1.48) 9.51 (1.67) 9.50 (1.54)	0.26 *p* = 0.771
Educational level				
Primary or below Secondary Tertiary/College or above	6.48 (1.92) 7.54 (2.35) 7.12 (2.57)	5.89 0.003**	7.00 (3.74) 9.09 (1.62) 9.73 (1.47)	15.07 <0.001***
Working status				
Full time Part time Nil ✡	7.27 (2.53) 6.83 (2.45) 6.95 (2.21)	1.18 0.309	9.73 (1.43) 9.59 (1.60) 9.28 (1.74)	5.14 0.006 **
Comorbid illness				
No Yes	7.21 (2.44) 6.70 (2.24)	0.25 0.08	9.61 (1.49) 9.28 (1.90)	1.73 0.054
Receive influenza vaccination over the past 12 months				
No Yes	7.28 (2.39) 5.31 (1.79)	5.19 < 0.001 ***	9.43 (1.50) 9.97 (1.80)	0.63 0.001 **
Would you perform hand hygiene more often during infectious disease outbreaks?				
No Yes	6.22 (2.32) 8.26 (2.06)	5.75 < 0.001 ***	9.44 (1.58) 9.58 (1.58)	0.11 0.377
How much time do you usually lather your hands with soap before rinsing?				
Less than 5 s 5–10 s 11–19 s 20 secs or more	5.65 (2.13) 8.06 (2.03) 8.09 (2.08) 8.20 (2.05)	65.05 < 0.001 ***	8.90 (1.72) 9.47 (1.34) 9.90 (2.03) 10.38 (1.41)	13.29 < 0.001 ***
Rub hands on own clothing				
Always/Sometimes Never	7.23 (2.40) 5.74 (2.08)	1.32 < 0.001 ***	9.52 (1.60) 9.56 (1.56)	0.35 0.736

Note: ▲—Independent *t*-test or one-way analysis of variance as appropriate; *—Statistically significant at *p* < 0.05; **—Statistically significant at *p* < 0.01; ***—Statistically significant at *p* < 0.001. s—second; ✡—retired/unemployed/housewife/student/voluntary job.

**Table 4 ijerph-17-04019-t004:** Self-reported hand cleaning and hand drying practice.

	Self-Reported Practice	Nepalese Residing in Hong Kong (*n* = 482)	Native Chinese Population of Hong Kong (*n* = 526)	Test Statistics & *p*-Value
Item		*n* (%)	*n* (%)	
	Hand cleaning			
1	Before handling food or cooking			
	No Water only Water and soap ABHR Wet wipes Not applicable ※	19 (4.2) 125 (26.1) 297 (61.7) 32 (6.8) 5 (1.2) 4	11 (2.1) 240 (46.0) 267 (51.1) 2 (0.4) 2 (0.4) 4	< 0.001 ◈ ***
2	After handling food or cooking			
	No Water only Water and soap ABHR Wet wipes Not applicable ※	78 (16.2) 112 (23.3) 256 (53.3) 32 (6.7) 4 (0.5) 0	9 (1.7) 160 (30.8) 349 (67.3) 1 (0.2) 0 (0.0) 7	< 0.001 ◈ ***
3	Before eating			
	No Water only Water and soap ABHR Wet wipes	64 (13.3) 127 (26.3) 250 (51.9) 38 (7.9) 3 (0.6)	58 (11.0) 227 (43.2) 223 (42.4) 6 (1.1) 12 (2.3)	< 0.001 ◈ ***
4	After urination			
	No Water only Water and soap ABHR Wet wipes	115 (23.9) 124 (25.7) 199 (41.3) 39 (8.1) 5 (1.0)	3 (0.6) 301 (57.2) 222 (42.2) 0 (0.0) 0 (0.0)	< 0.001 ◈ ***
5	After defecation			
	No Water only Water and soap ABHR Wet wipes	66 (13.7) 27 (5.6) 317 (65.8) 69 (14.3) 3 (0.6)	1 (0.2) 90 (17.1) 434 (82.5) 1 (0.2) 0 (0.0)	< 0.001 ◈ ***
6	After feeding a child			
	No Water only Water and soap ABHR Wet wipes Not applicable ※	55 (12.4) 127 (28.5) 216 (48.5) 40 (9.1) 6 (1.5) 38	189 (35.9) 159 (30.2) 171 (32.5) 2 (0.4) 5 (1.0) 0	< 0.001 ◈ ***
7	After caring for a sick person			
	No Water only Water and soap ABHR Wet wipes Not applicable ※	26 (5.5) 79 (16.9) 262 (56.0) 100 (21.4) 2 (0.2) 13	4 (0.9) 59 (13.1) 304 (67.7) 80 (17.9) 2 (0.4) 77	< 0.001 ◈ ***
8	After daily work			
	No Water only Water and soap ABHR Wet wipes Not applicable ※	14 (3.3) 79 (19.0) 264 (63.0) 55 (13.3) 6 (1.4) 64	70 (13.9) 175 (34.6) 243 (48.1) 12 (2.4) 5 (1.0) 21	< 0.001 ◈ ***
9	When your hands are visibly dirty			
	No Water only Water and soap ABHR Wet wipes	75 (15.6) 90 (18.7) 260 (53.8) 50 (10.4) 7 (1.5)	3 (0.6) 11 (2.0) 505 (96.0) 2 (0.4) 5 (1.0)	< 0.001 ◈ ***
10	After sneezing or coughing			
	No Water only Water and soap ABHR Wet wipes	126 (26.1) 80 (16.6) 212 (44.0) 57 (11.8) 7 (1.5)	104 (19.8) 146 (27.8) 195 (37.0) 34 (6.5) 47 (8.9)	< 0.001 ◈ ***
11	After touching livestock(s)			
	No Water only Water and soap ABHR Wet wipes Not applicable ※	9 (2.4) 53 (14.0) 244 (64.5) 69 (18.3) 3 (0.8) 104	2 (0.6) 25 (7.6) 275 (83.0) 24 (7.3) 5 (1.5) 195	< 0.001 ◈ ***
12	After touching animal(s) waste			
	No Water only Water and soap ABHR Wet wipes Not applicable ※	25 (6.2) 57 (14.1) 264 (64.8) 60 (14.9) 0 (0.0) 76	0 (0.0) 11 (3.1) 318 (90.2) 21 (5.9) 3 (0.8) 173	< 0.001 ◈ ***
13	After disposal of garbage bag			
	No Water only Water and soap ABHR Wet wipes	203 (42.1) 53 (11.0) 185 (38.4) 40 (8.3) 1 (0.2)	16 (3.0) 148 (28.1) 351 (66.8) 10 (1.9) 1 (0.2)	< 0.001 ◈ ***
14	After gardening			
	No Water only Water and soap ABHR Not applicable ※	28 (7.5) 43 (11.5) 271 (72.3) 33 (8.7) 107	5 (1.4) 105 (28.5) 257 (69.8) 1 (0.3) 158	< 0.001 ◈ ***
15	Would you perform hand hygiene more often during infectious disease outbreaks?			
	No Yes	261 (54.1) 221 (45.9)	121 (23.0) 405 (77.0)	< 0.001 ★ ***
16	How much time do you usually lather your hands with soap before rinsing?			
	Less than 5 s 5–10 s 11–19 s 20 s or more	197 (40.9) 78 (16.2) 70 (14.5) 137 (28.4)	86 (16.3) 303 (57.6) 72 (13.7) 65 (12.4)	< 0.001 ★ ***
17	Ignore handwashing if in a hurry			
	Always Sometimes Never	52 (10.8) 287 (59.5) 143 (29.7)	5 (1.0) 53 (10.0) 467 (89.0)	< 0.001 ★ ***
18	Ignore handwashing when nobody in the washroom			
	Always Sometimes Never	20 (4.1) 270 (56.0) 192 (39.9)	1 (0.2) 30 (5.7) 494 (94.1)	< 0.001 ◈ ***
19	Ignore handwashing if I have only urinated			
	Always Sometimes Never	27 (5.6) 295 (61.2) 160 (33.2)	6 (1.1) 60 (11.4) 459 (87.5)	< 0.001 ★ ***
	Hand drying			
20	Rub hands on own clothing			
	Always Sometimes Never	367 (76.1) 80 (16.6) 35 (7.3)	40 (7.6) 184 (35.0) 301 (57.4)	< 0.001 ★ ***
21	Air evaporation			
	Always Sometimes Never	125 (25.9) 215 (44.6) 142 (29.5)	99 (18.8) 265 (50.5) 161 (30.7)	< 0.001 ★ ***
22	Use personal towel or handkerchief			
	Always Sometimes Never	160 (33.2) 240 (49.8) 82 (17.0)	53 (10.1) 135 (25.7) 337 (64.2)	< 0.001 ★ ***
23	Use own disposable tissue			
	Always Sometimes Never	139 (28.8) 222 (46.0) 121 (25.2)	188 (35.8) 240 (45.7) 97 (18.5)	0.033 ★ *
24	Paper towels supplied by the washroom			
	Always Sometimes Never	118 (24.5) 199 (41.3) 165 (34.2)	364 (69.3) 144 (27.5) 17 (3.2)	< 0.001 ★ ***
25	Shaking my hands to get rid of excess water before drying			
	Always Sometimes Never	277 (57.5) 180 (37.2) 25 (5.3)	279 (53.1) 184 (35.0) 62 (11.9)	0.005 ★ **
26	How many paper towels do you commonly used			
	One Two Three Four or more Not applicable ※	148 (33.4) 217 (48.9) 57 (12.9) 21 (4.8) 39	240 (46.3) 234 (45.2) 37 (7.1) 7 (1.4) 7	< 0.001 ★ ***
27	Warm hand dryer			
	Always Sometimes never	178 (36.9) 231 (48.0) 73 (15.1)	105 (20.0) 334 (63.6) 86 (16.4)	< 0.001 ★ ***
28	Jet hand dryer			
	Always Sometimes Never	111 (23.1) 285 (59.2) 86 (17.7)	98 (18.7) 319 (60.7) 108 (20.6)	0.236 ★
29	If warm hand dryer is used, how do you usually position your hands?			
	Rubbing the hands during drying Hold the hands stationary during drying Not applicable ※	164 (46.6) 188 (53.4) 130	247 (52.7) 222 (47.3) 56	0.120 ★
30	Average time for using warm hand dryer (in seconds)			
	Less than 5 sec 5–10 s 11–20 s 21–30 s 31–40 s 41 s or more Not applicable ※	55 (12.0) 175 (37.8) 152 (32.9) 53 (11.4) 14 (3.1) 13 (2.8) 20	40 (8.5) 243 (51.8) 141 (30.1) 29 (6.2) 11 (2.3) 5 (1.1) 56	0.001 ★ **
31	Average time for using jet hand dryer (in seconds)			
	Less than 5 s 5–10 s 11–20 s 21–30 s 31–40 s 41 s or more Not applicable ※	44 (10.0) 149 (34.0) 147 (33.7) 44 (10.0) 38 (8.7) 16 (3.6) 44	53 (12.2) 226 (51.8) 120 (27.5) 19 (4.4) 13 (3.0) 5 (1.1) 89	< 0.001 ★ ***
32	Methods commonly used for hand drying at home			
	Rubbed hands on own clothes Air evaporation Use own towel A towel shared with family members Paper towels Warm hand dryer Jet hand dryer	118 (24.3) 114 (23.6) 160 (33.1) 30 (6.2) 50 (10.4) 7 (1.5) 3 (0.9)	21 (4.0) 25 (4.7) 171 (32.5) 236 (45.0) 68 (13.0) 3 (0.6) 1 (0.2)	< 0.001 ◈ ***

Note: ABHR—Alcohol based hand rub; ◈—Fisher’s exact test; ★—Chi-square test; *—Statistically significant at *p* < 0.05; **—Statistically significant at *p* < 0.01; ***—Statistically significant at *p* < 0.001; ※—Not applicable cases were excluded from percentage calculation and the analyses. s: seconds

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
