# Peer review of "Knowledge Level and Hand Hygiene Practice of Nepalese Immigrants and Their Host Country Population: A Comparative Study"

_ijerph, 2020, doi:10.3390/ijerph17114019_

Round 1

Reviewer 1 Report

The work is based on a solid experimental design and on a well-conducted statistical processing. The text describes the work completely. I only ask you to make corrections on the following two points of the text:

61 The bibliographic reference [12] is linked to a paper whose contents do not seem at all consistent with the statement made. Please, verify and correct.

156 The specific reference to the age category (18-19) would not be consistent with the statistical data. It is correct to indicate that there are significant age-related differences, as well as that means indicate the worst levels of knowledge in young and elder people, but there is no statistical analyses that revealed significant differences between the age categories. I ask to correct the statement.

Reviewer 2 Report

General comments

The reported research is interesting and relevant. As it is a sensitive topic reporting about ethnical groups and their educations and behavior the manuscript should be rechecked if some sentences could be misunderstood. One example are the column headers of table 1. One column is about the Nepalese resisting in Hong Kong and the other one the Hong Kong population. If these people live there, they are a part of Hong Kongs population as well. There were other parts of the manuscript in which it sounds as the immigrants have not immigrated yet.

HH is an important contribution to public health – no doubts but exists there any statistics about how the Nepalese impact on the current infectious disease level is or with other words – is there a measurable impact on public health due to the poor HH habits of the Nepalese?

Specific Comments:

Line 30: Do you mean “lack of knowledge or poor knowledge”?

Line 32: This is a general fact – not only in low-income countries.

Line 44: developed is OK but it sounds like a positive progress – I would write: “HH practices which are the common standard…”

Line 64/65: “…they have rarely been studied…” – rewrite this part. “Less is known” sounds more positive as an ethnicity which is not studied.

Line 99: Is this only the case for the online based survey – not for the paper based one?

Line 122: Rewrite the sentence as Nepalese are a part of the local population as well if they live there.

Line 125: In science – especially if you compare 11 min and 16 min the time must be measured, or it must be deleted. Estimating a time of a survey is not acceptable in this context.

Line 132: Please explain the connection between influence vaccination and HH.

Table 1: Adjust the column headers and check if the % sum in the penultimate row is correct (=100,1%)

Table 2: Was the abbreviation explained in the survey? Is not expected that non-healthcare people understand what URTI or ILI is.

Table 2: The abbreviations URTI/ILI were not introduced in the manuscript and must be written in the text.

Table 2: What are the statistics about? Have you a p value for yes/no? This will always be 0.0

Table 2 rows 10 and 11: Adjust the position of the results. They dropped one row below.

Table 2 last row: This is the sum of all correct answers. How to calculate a SD from a sum?

Line 162: Add the interval of the level (0-12?).

Table 3 row “Influenza vaccination”: As I see no connection between HH and Influence vaccination or the other studied parameters like education level ad so on, this row contributes no information and can be deleted.

Table 4: All numbers in the 1st column >10 are not completely visible. Adjust the size of the columns.

References: Check the varying use of the DOI in the reference section.

Reviewer 3 Report

Dear Editor,

Regarding the manuscript "Knowledge level and hand hygiene practice of Nepalese immigrants and their host country population: A comparative study", I have finished the reading and I judge it is a good piece of work.

The manuscript has been written in a very clear way, being straighfoward for readers. I judge it has a good potential to be citable in a near future.

As minor suggestions:

  • The acronym for seconds in International System of Units (SI) is s, not sec.
  • L116: please remove the word "appropriate".
  • L119: p<=0.05 instead p<0.05.
  • L304 "SurveyMonkey".

Congratulations.

Round 2

Reviewer 2 Report

Thank you for providing the revised version of your manuscript. I have no further suggestions for an improvement.